# On the Sublinear Regret of GP-UCB

**Justin Whitehouse**
Carnegie Mellon University
jwhiteho@andrew.cmu.edu

**Zhiwei Steven Wu**
Carnegie Mellon University
zstevenwu@cmu.edu

**Aaditya Ramdas**
Carnegie Mellon University
aramdas@cmu.edu

## Abstract

In the kernelized bandit problem, a learner aims to sequentially compute the optimum of a function lying in a reproducing kernel Hilbert space given only noisy evaluations at sequentially chosen points. In particular, the learner aims to minimize regret, which is a measure of the suboptimality of the choices made. Arguably the most popular algorithm is the Gaussian Process Upper Confidence Bound (GP-UCB) algorithm, which involves acting based on a simple linear estimator of the unknown function. Despite its popularity, existing analyses of GP-UCB give a suboptimal regret rate, which fails to be sublinear for many commonly used kernels such as the Matérn kernel. This has led to a longstanding open question: are existing regret analyses for GP-UCB tight, or can bounds be improved by using more sophisticated analytical techniques? In this work, we resolve this open question and show that GP-UCB enjoys nearly optimal regret. In particular, our results yield sublinear regret rates for the Matérn kernel, improving over the state-of-the-art analyses and partially resolving a COLT open problem posed by Vakili et al. Our improvements rely on a key technical contribution — regularizing kernel ridge estimators in proportion to the smoothness of the underlying kernel $k$. Applying this key idea together with a largely overlooked concentration result in separable Hilbert spaces (for which we provide an independent, simplified derivation), we are able to provide a tighter analysis of the GP-UCB algorithm.

## 1 Introduction

An essential problem in areas such as econometrics [12, 13], medicine [22, 23], optimal control [4, 3], and advertising [21] is to optimize an unknown function given *bandit feedback*, in which algorithms only get to observe the outcomes for the chosen actions. Due to the bandit feedback, there is a fundamental tradeoff between *exploiting* what has been observed about the local behavior of the function and *exploring* to learn more about the function's global behavior. There has been a long line of work on bandit learning that investigates this tradeoff across different settings, including multi-armed bandits [29, 19, 37], linear bandits [2, 30], and kernelized bandits [5, 26, 32].

In this work, we focus on the kernelized bandit framework, which can be viewed as an extension of the well-studied linear bandit setting to an infinite-dimensional reproducing kernel Hilbert space (or RKHS) $(H, \langle \cdot, \cdot \rangle_H)$. In this problem, there is some unknown function $f^* : \mathcal{X} \to \mathbb{R}$ of bounded norm in $H$, where $\mathcal{X} \subset \mathbb{R}^d$ is a bounded set. In each round $t \in [T]$, the learner uses previous observations to select an action $X_t \in \mathcal{X}$, and then observes feedback $Y_t := f^*(X_t) + \epsilon_t$, where $\epsilon_t$ is a zero-mean noise variable. The learner aims to minimize (with high probability) the regret at time

37th Conference on Neural Information Processing Systems (NeurIPS 2023).

$T$, which is defined as

$$R_T := \sum_{t=1}^{T} f^*(x^*) - f^*(X_t)$$

where $x^* := \arg\max_{x \in \mathcal{X}} f^*(x)$. The goal is to develop simple, efficient algorithms for the kernelized bandit problem that minimize regret $R_T$. We make the following standard assumption. We also make assumptions on the underlying kernel $k$, which we discuss in Section 2.

**Assumption 1.** *We assume that (a) there is some constant $D > 0$ known to the learner such that $\|f^*\|_H \leq D$ and (b) for every $t \geq 1$, $\epsilon_t$ is $\sigma$-subGaussian conditioned on $\sigma(Y_{1:t-1}, X_{1:t})$.*

Arguably the simplest algorithm for the kernelized bandit problem is GP-UCB (Gaussian process upper confidence bound) [31, 5]. GP-UCB works by maintaining a kernel ridge regression estimator of the unknown function $f^*$ alongside a confidence ellipsoid, optimistically selecting in each round the action that provides the maximal payoff over all feasible functions. Not only is GP-UCB efficiently computable thanks to the kernel trick, but it also offers strong empirical guarantees [5]. The only seeming deficit of GP-UCB is its regret guarantee, as existing analyses only show that, with high probability, $R_T = \widetilde{O}(\gamma_T \sqrt{T})$, where $\gamma_T$ is a kernel-dependent measure of complexity known as the maximum information gain [31, 6]. In contrast, more complicated, less computationally efficient algorithms such as SupKernelUCB [34, 25] have been shown to obtain regret bounds of $\widetilde{O}(\sqrt{\gamma_T T})$, improving over the analysis of GP-UCB by a multiplicative factor of $\sqrt{\gamma_T}$. This gap is stark as the bound $\widetilde{O}(\gamma_T \sqrt{T})$ fails, in general, to be sub-linear for the practically relevant Matérn kernel, whereas $\widetilde{O}(\sqrt{\gamma_T T})$ is sublinear for *any* kernel experiencing polynomial eigendecay [32].

This discrepancy has prompted the development of many variants of GP-UCB that, while less computationally efficient, offer better regret guarantees in some situations [17, 27, 28]. (See a detailed discussion of these algorithms along with other related work in Appendix A.) However, the following question remains an open problem in online learning [33]: are existing analyses of vanilla GP-UCB tight, or can an improved analysis show GP-UCB enjoys sublinear regret?

## 1.1 Contributions

In this work, we show that GP-UCB obtains almost optimal, sublinear regret for any kernel experiencing polynomial eigendecay. This, in particular, implies that GP-UCB obtains sublinear regret for the commonly used Matérn family of kernels. We provide a brief roadmap of our paper below.

1. In Section 3, we provide background into self-normalized concentration in Hilbert spaces. In particular, in Theorem 1, we provide an independent, simplified derivation of a bound due to Abbasi-Yadkori [1], which concerns to self-normalized concentration of certain process in separable Hilbert spaces. This bound has been largely overlooked in the kernel bandit literature, so we draw attention to it in hopes it can be leveraged in solving further kernel-based learning problems. As opposed to the existing bound of Chowdhury and Gopalan [5], which involves employing a complicated "double mixture" argument, the bound we present follows directly from applying the well-studied finite-dimensional method of mixtures alongside a simple truncation argument [7–9, 2]. These bounds are clean and show simple dependence on the regularization parameter.

2. In Section 4, we use leverage the self-normalized concentration detailed in Theorem 1 to provide an improved regret analysis for GP-UCB. By carefully choosing regularization parameters based on the smoothness of the underlying kernel, we demonstrate that GP-UCB enjoys sublinear regret of $\widetilde{O}\left(T^{\frac{3+\beta}{2+2\beta}}\right)$ for any kernel experiencing $(C, \beta)$-polynomial eigendecay. As a special case of this result, we obtain regret bounds of $\widetilde{O}\left(T^{\frac{\nu+2d}{2\nu+2d}}\right)$ for the commonly used Matérn kernel with smoothness $\nu$ in dimension $d$. Our new analysis improves over existing state-of-the-art analysis for GP-UCB, which fails to guarantee sublinear regret in general for the Matérn kernel family [5], and thus partially resolves an open problem posed by [33] on the suboptimality of GP-UCB.

In sum, our results show that GP-UCB, the go-to algorithm for the kernelized bandit problem, is nearly optimal, coming close to the algorithm-independent lower bounds of Scarlett et al. [25]. Our

work thus can be seen as providing theoretical justification for the strong empirical performance of GP-UCB [31]. Perhaps the most important message of our work is the importance of careful regularization in online learning problems. While many existing bandit works treat the regularization parameter as a small, kernel-independent constant, we are able to obtain significant improvements by carefully selecting the regularization parameter. We hope our work will encourage others to pay close attention to the selection of regularization parameters in future works.

## 2 Background and Problem Statement

**Notation.** We briefly touch on basic definitions and notational conveniences that will be used throughout our work. If $a_1, \ldots, a_t \in \mathbb{R}$, we let $a_{1:t} := (a_1, \ldots, a_t)^\top$. Let $(H, \langle \cdot, \cdot \rangle_H)$ be a reproducing kernel Hilbert space associated with a kernel $k : \mathcal{X} \times \mathcal{X} \to \mathbb{R}$. We refer to the identity operator on $H$ as $\mathrm{id}_H$. This is distinct from the identity mapping on $\mathbb{R}^d$, which we will refer to as $I_d$. For elements $f, g \in H$, we define their outer product as $fg^\top := f\langle g, \cdot \rangle_H$ and inner product as $f^\top g := \langle f, g \rangle_H$. For any $t \geq 1$ and sequence of points $x_1, \ldots, x_t \in \mathcal{X}$ (which will typically be understood from context), let $\Phi_t := (k(\cdot, x_1), \ldots, k(\cdot, x_t))^\top$. We can respectively define the Gram matrix $K_t : \mathbb{R}^t \to \mathbb{R}^t$ and covariance operator $V_t : H \to H$ as $K_t := (k(x_i, x_j))_{i,j \in [t]} = \Phi_t \Phi_t^\top$ and $V_t := \sum_{s=1}^t k(\cdot, x_s) k(\cdot, x_s)^\top = \Phi_t^\top \Phi_t$. These two operators essentially encode the same information about the observed data points, the former being easier to work with when actually performing computations (by use of the well known kernel trick) and latter being easier to algebraically manipulate.

Suppose $A : H \to H$ is a Hermitian operator of finite rank; enumerate its non-zero eigenvalues as $\lambda_1(A), \ldots, \lambda_k(A)$. We can define the Fredholm determinant of $I + A$ as $\det(I + A) := \prod_{m=1}^k (1 + \lambda_i(A))$ [20]. For any $t \geq 1, \rho > 0$, and $x_1, \ldots, x_t \in \mathcal{X}$, one can check via a straightforward computation that $\det(I_t + \rho^{-1} K_t) = \det(\mathrm{id}_H + \rho^{-1} V_t)$, where $K_t$ and $V_t$ are the Gram matrix and covariance operator defined above. We, again, will use these two quantities interchangeably in the sequel, but will typically prefer the latter in our proofs.

If $(H, \langle \cdot, \cdot \rangle_H)$ is a (now general) separable Hilbert space and $(\varphi_n)_{n \geq 1}$ is an orthonormal basis for $H$, for any $N \geq 1$ we can define the orthogonal projection operator $\pi_N : H \to \mathrm{span}\{\varphi_1, \ldots, \varphi_N\} \subset H$ by $\pi_N f := \sum_{n=1}^N \langle f, \varphi_n \rangle_H \varphi_n$. We can correspondingly the define the projection onto the remaining basis functions to be the map $\pi_N^\perp : H \to \mathrm{span}\{\varphi_1, \ldots, \varphi_N\}^\perp$ given by $\pi_N^\perp f := f - \pi_N f$. Lastly, if $A : H \to H$ is a symmetric, bounded linear operator, we let $\lambda_{\max}(A)$ denote the maximal eigenvalue of $A$, when such a value exists. In particular, $\lambda_{\max}(A)$ will exist whenever $A$ has a finite rank, as will typically be the case considered in this paper.

**Basics on RKHSs.** Let $\mathcal{X} \subset \mathbb{R}^d$ be some domain. A *kernel* is a positive semidefinite map $k : \mathcal{X} \times \mathcal{X} \to \mathbb{R}$ that is square-integrable, i.e. $\int_{\mathcal{X}} \int_{\mathcal{X}} |k(x, y)|^2 dx dy < \infty$. Any kernel $k$ has an associated *reproducing kernel Hilbert space* or *RKHS* $(H, \langle \cdot, \cdot \rangle_H)$ containing the closed span of all partial kernel evaluations $k(\cdot, x), x \in \mathcal{X}$. In particular, the inner product $\langle \cdot, \cdot \rangle_H$ on $H$ satisfies the reproducing relationship $f(x) = \langle f, k(\cdot, x) \rangle_H$ for all $x \in \mathcal{X}$.

A kernel $k$ can be associated with a corresponding *Hilbert-Schmidt operator*, which is the Hermitian operator $T_k : L^2(\mathcal{X}) \to L^2(\mathcal{X})$ given by $(T_k f)(x) := \int_{\mathcal{X}} f(y) k(x, y) dy$ for any $x \in \mathcal{X}$. In short, $T_k$ can be thought of as "smoothing out" or "mollifying" a function $f$ according to the similarity metric induced by $k$. $T_k$ plays a key role in kernelized learning through *Mercer's Theorem*, which gives an explicit representation for $H$ in terms of the eigenvalues and eigenfunctions of $T_k$.

**Fact 1** (**Mercer's Theorem**). *Let $(H, \langle \cdot, \cdot \rangle_H)$ be the RKHS associated with kernel $k$, and let $(\mu_n)_{n \geq 1}$ and $(\phi_n)_{n \geq 1}$ be the sequence of non-increasing eigenvalues and corresponding eigenfunctions for $T_k$. Let $(\varphi_n)_{n \geq 1}$ be the sequence of rescaled functions $\varphi_n := \sqrt{\mu}_n \phi_n$. Then,*

$$H = \left\{ \sum_{n=1}^\infty \theta_n \varphi_n : \sum_{n=1}^\infty \theta_n^2 < \infty \right\},$$

*and $(\varphi_n)_{n \geq 1}$ forms an orthonormal basis for $(H, \langle \cdot, \cdot \rangle_H)$.*

We make the following assumption throughout the remainder of our work, which is standard and comes from Vakili et al. [32].

**Assumption 2 (Assumption on kernel $k$).** *The kernel $k : \mathcal{X} \times \mathcal{X} \to \mathbb{R}$ satisfies (a) $|k(x, y)| \leq L$ for all $x, y \in \mathcal{X}$, for some constant $L > 0$ and (b) $|\phi_n(x)| \leq B$ for all $x \in \mathcal{X}$, for some $B > 0$.*

**"Complexity" of RKHS's.** By the eigendecay of a kernel $k$, we really mean the rate of decay of the sequence of eigenvalues $(\mu_n)_{n \geq 1}$. In the literature, there are two common paradigms for studying the eigendecay of $k$: $(C_1, C_2, \beta)$-exponential eigendecay, under which $\forall n \geq 1, \mu_n \leq C_1 \exp(-C_2 n^\beta)$, and $(C, \beta)$-polynomial eigendecay, under which $\forall n \geq 1, \mu_n \leq Cn^{-\beta}$. For kernels experiencing exponential eigendecay, of which the squared exponential is the most important example, GP-UCB is known to be optimal up to poly-logarithmic factors. However, for kernels experiencing polynomial eigendecay, of which the Matérn family is a common example, existing analyses of GP-UCB fail to yield sublinear regret. It is this latter case we focus on in this work.

Given the above representation in Fact 1, it is clear that the eigendecay of the kernel $k$ governs the "complexity" or "size" of the RKHS $H$. We make this notion of complexity precise by discussing *maximum information gain*, a sequential, kernel-dependent quantity governing concentration and hardness of learning in RKHS's [6, 31, 32].

Let $t \geq 1$ and $\rho > 0$ be arbitrary. The maximum information gain at time $t$ with regularization $\rho$ is the scalar $\gamma_t(\rho)$ given by

$$\gamma_t(\rho) := \sup_{x_1, \ldots, x_t \in \mathcal{X}} \frac{1}{2} \log \det \left( \mathrm{id}_H + \rho^{-1} V_t \right) = \sup_{x_1, \ldots, x_t \in \mathcal{X}} \frac{1}{2} \log \det \left( I_t + \rho^{-1} K_t \right).$$

Our presentation of maximum information gain differs from some previous works in that we encode the regularization parameter $\rho$ into our notation. This inclusion is key for our results, as we obtain improvements by carefully selecting $\rho$. Vakili et al. [32] bound the rate of growth of $\gamma_t(\rho)$ in terms of the rate of eigendecay of the kernel $k$. We leverage the following fact in our main results.

**Fact 2 (Corollary 1 in Vakili et al. [32]).** *Suppose that kernel $k$ satisfies Assumption 2 and experiences $(C, \beta)$-polynomial eigendecay. Then, for any $t \geq 1$, we have*

$$\gamma_t(\rho) \leq \left( \left( \frac{CB^2 t}{\rho} \right)^{1/\beta} \log^{-1/\beta} \left( 1 + \frac{Lt}{\rho} \right) + 1 \right) \log \left( 1 + \frac{Lt}{\rho} \right).$$

We last define the practically relevant Matérn kernel and discuss its eigendecay.

**Definition/Fact 3.** The Matérn kernel with bandwidth $\sigma > 0$ and smoothness $\nu > 1/2$ is given by

$$k_{\nu, \sigma}(x, y) := \frac{1}{\Gamma(\nu) 2^{\nu-1}} \left( \frac{\sqrt{2\nu} \|x - y\|_2}{\sigma} \right)^\nu B_\nu \left( \frac{\sqrt{2\nu} \|x - y\|_2}{\sigma} \right),$$

where $\Gamma$ is the gamma function and $B_\nu$ is the modified Bessel function of the second kind. It is known that there is some constant $C > 0$ that may depend on $\sigma$ but not on $d$ or $\nu$ such that $k_{\nu,\sigma}$ experiences $\left( C, \frac{2\nu+d}{d} \right)$-eigendecay [24, 32].

**Basics on martingale concentration:** A filtration $(\mathcal{F}_t)_{t \geq 0}$ is a sequence of $\sigma$-algebras satisfying $\mathcal{F}_t \subset \mathcal{F}_{t+1}$ for all $t \geq 1$. If $(M_t)_{t \geq 0}$ is a $H$-valued process, we say $(M_t)_{t \geq 0}$ is a martingale with respect to $(\mathcal{F}_t)_{t \geq 0}$ if (a) $(M_t)_{t \geq 0}$ is $(\mathcal{F}_t)_{t \geq 0}$-adapted, and (b) $\mathbb{E}(M_t \mid \mathcal{F}_{t-1}) = M_{t-1}$ for all $t \geq 1$. An $\mathbb{R}$-valued process is called a supermartingale if the equality in (b) is replaced with "$\leq$", i.e. supermartingales tend to decrease. Martingales are useful in many statistical applications due to their strong concentration of measure properties [15, 36]. The follow fact can be leveraged to provide time-uniform bounds on the growth of any non-negative supermartingale.

**Fact 4 (Ville's Inequality).** *Let $(M_t)_{t \geq 0}$ be a non-negative supermartingale with respect to some filtration. Suppose $\mathbb{E}M_0 = 1$. Then, for any $\delta \in (0, 1)$, we have*

$$\mathbb{P} \left( \exists t \geq 0 : M_t \geq \frac{1}{\delta} \right) \leq \delta.$$

See Howard et al. [14] for a self-contained proof of Ville's inequality, and many applications.

If $\mathcal{F}$ is a $\sigma$-algebra, and $\epsilon$ is an $\mathbb{R}$-valued random variable, we say $\epsilon$ is $\sigma$-subGaussian conditioned on $\mathcal{F}$ if, for any $\lambda \in \mathbb{R}$, we have $\log \mathbb{E} \left( e^{\lambda \epsilon} \mid \mathcal{F} \right) \leq \frac{\lambda^2 \sigma^2}{2}$; in particular this condition implies

that $\epsilon$ is mean zero. With this, we state the following result on self-normalized processes. To our understanding, the following result was first presented in some form as Example 4.2 of de la Peña et al. [8] (in the setting of continuous local martingales), and can be derived leveraging the argument of Theorem 1 in de la Peña et al. [9]. The exact form below was established (in the setting of discrete-time processes) in Theorem 1 of Abbasi-Yadkori et al. [2], which is commonly leveraged to construct confidence ellipsoids in the linear bandit setting.

**Fact 5** (**Example 4.2 from [8], Theorem 1 from [2]**). *Let $(\mathcal{F}_t)_{t \geq 0}$ be a filtration, let $(X_t)_{t \geq 1}$ be an $(\mathcal{F}_t)_{t \geq 0}$-predictable sequence in $\mathbb{R}^d$, and let $(\epsilon_t)_{t \geq 1}$ be a real-valued $(\mathcal{F}_t)_{t \geq 1}$-adapted sequence such that conditional on $\mathcal{F}_{t-1}$, $\epsilon_t$ is mean zero and $\sigma$-subGaussian. Then, for any $\rho > 0$, the process $(M_t)_{t \geq 0}$ given by*

$$M_t := \frac{1}{\sqrt{\det(I_d + \rho^{-1}V_t)}} \exp\left\{\frac{1}{2}\left\|(\rho I_d + V_t)^{-1/2}S_t/\sigma\right\|_2^2\right\}$$

*is a non-negative supermartingale with respect to $(\mathcal{F}_t)_{t \geq 0}$, where $S_t := \sum_{s=1}^t \epsilon_s X_s$ and $V_t := \sum_{s=1}^t X_s X_s^\top$. Consequently, by Fact 4, for any confidence $\delta \in (0, 1)$, the following holds: with probability at least $1 - \delta$, simultaneously for all $t \geq 1$, we have*

$$\left\|(V_t + \rho I_d)^{-1/2}S_t\right\|_2 \leq \sigma\sqrt{2\log\left(\frac{1}{\delta}\sqrt{\det(I_d + \rho^{-1}V_t)}\right)}.$$

Note the simple dependence on the regularization parameter $\rho > 0$ in the above bound. While the regularization parameter $\rho$ doesn't prove important in regret analysis for linear bandits (where $\rho$ is treated as constant), the choice for $\rho$ will be critical in our setting. In the following section, we will discuss how Fact 5 can be extended to the setting of separable Hilbert spaces essentially verbatim (an observation first noticed by Abbasi-Yadkori [1]).

## 3 A Remark on Self-Normalized Concentration in Hillbert Spaces

We begin by discussing a key, self-normalized concentration inequality for martingales. We use this bound in the sequel to construct simpler, more flexible confidence ellipsoids than currently exist for GP-UCB. The bound we present (in Theorem 1 below) is, more or less, equivalent to Corollary 3.5 in the thesis of Abbasi-Yadkori [1]. Our result is mildly more general in the sense that it directly argues that a target mixture process is a nonnegative supermartingale. The result in Abbasi-Yadkori [1] is more general in the sense it allows the regularization (or shift) matrix to be non-diagonal. Either concentration result is sufficient for the regret bounds obtained in the sequel.

The aforementioned corollary in [1], quite surprisingly, has not been referenced in central works on the kernelized bandit problem, namely Chowdhury and Gopalan [5] and Vakili et al. [32, 33]. In fact, strictly weaker versions of the conclusion have been independently rediscovered in the context of kernel regression [10]. We emphasize that this result of Abbasi-Yadkori [1] (and the surrounding technical conclusions) are very general and may allow for further improvements in problems related to kernelized learning.

We now present Theorem 1, providing a brief sketch and a full proof in Appendix B. We believe our proof, which directly shows a target process is a nonnegative supermartingale, is of independent interest when compared to that of Abbasi-Yadkori [1] due to its simplicity. In particular, our proof follows from first principles, avoiding advanced topological notions of convergence (e.g. in the weak operator topology) and existence of certain Gaussian measures on separable Hilbert spaces, which were heavily utilized in the proof of Corollary 3.5 in Abbasi-Yadkori [1].

**Theorem 1** (**Self-normalized concentration in Hilbert spaces**). *Let $(\mathcal{F}_t)_{t \geq 0}$ be a filtration, $(f_t)_{t \geq 1}$ be an $(\mathcal{F}_t)_{t \geq 0}$-predictable sequence in a separable Hilbert space[1] $H$ such that $\|f_t\|_H < \infty$ a.s. for all $t \geq 0$, and $(\epsilon_t)_{t \geq 1}$ be an $(\mathcal{F}_t)_{t \geq 1}$-adapted sequence in $\mathbb{R}$ such that conditioned on $\mathcal{F}_{t-1}$, $\epsilon_t$ is mean zero and $\sigma$-subGaussian. Defining $S_t := \sum_{s=1}^t \epsilon_s f_s$ and $V_t := \sum_{s=1}^t f_s f_s^\top$, we have that for any $\rho > 0$, the process $(M_t)_{t \geq 0}$ defined by*

$$M_t := \frac{1}{\sqrt{\det(\mathrm{id}_H + \rho^{-1}V_t)}} \exp\left\{\frac{1}{2}\left\|(\rho\,\mathrm{id}_H + V_t)^{-1/2}S_t/\sigma\right\|_H^2\right\}$$

---

[1]A space is separable if it has a countable, dense set. Separability is key, because it means we have a countable basis, whose first $N$ elements we project onto.

*is a nonnegative supermartingale with respect to* $(\mathcal{F}_t)_{t \geq 0}$. *Consequently, by Fact 4, for any* $\delta \in (0, 1)$, *with probability at least* $1 - \delta$, *simultaneously for all* $t \geq 1$, *we have*

$$\left\| (V_t + \rho I_d)^{-1/2} S_t \right\|_H \leq \sigma \sqrt{2 \log \left( \frac{1}{\delta} \sqrt{\det(\mathrm{id}_H + \rho^{-1} V_t)} \right)}.$$

We can summarize our independent proof in two simple steps. First, following from Fact 5, the bound in Theorem 1 holds when we project $S_t$ and $V_t$ onto a finite number $N$ of coordinates, defining a "truncated" nonnegative supermartingale $M_t^{(N)}$. Secondly, we can make a limiting arugment, showing $M_t^{(N)}$ is "essentially" $M_t$ for large values of $N$.

*Proof Sketch for Theorem 1.* Let $(\varphi_n)_{n \geq 1}$ be an orthonormal basis for $H$, and, for any $N \geq 1$, let $\pi_N$ denote the projection operator onto $\bar{H}_N := \mathrm{span}\{\varphi_1, \ldots, \varphi_N\}$. Note that the projected process $(\pi_N S_t)_{t \geq 1}$ is an $H$-valued martingale with respect to $(\mathcal{F}_t)_{t \geq 0}$. Further, note that the projected variance process $(\pi_N V_t \pi_N^\top)_{t \geq 0}$ satisfies

$$\pi_N V_t \pi_N^\top = \sum_{s=1}^{t} (\pi_N f_s)(\pi_N f_s)^\top.$$

Since, for any $N \geq 1$, $H_N$ is a finite-dimensional Hilbert space, it follows from Lemma 1 that the process $(M_t^{(N)})_{t \geq 0}$ given by

$$M_t^{(N)} := \frac{1}{\sqrt{\det(\mathrm{id}_H + \rho^{-1} \pi_N V_t \pi_N^\top)}} \exp \left\{ \frac{1}{2} \left\| (\rho \, \mathrm{id}_H + \pi_N V_t \pi_N^\top)^{-1/2} \pi_N S_t \right\|_H^2 \right\},$$

is a nonnegative supermartingale with respect to $(\mathcal{F}_t)_{t \geq 0}$. One can check that, for any $t \geq 0$, $M_t^{(N)} \xrightarrow[N \to \infty]{} M_t$. Thus, Fatou's Lemma implies

$$
\begin{aligned}
\mathbb{E}\left(M_t \mid \mathcal{F}_{t-1}\right) &= \mathbb{E}\left(\liminf_{N \to \infty} M_t^{(N)} \mid \mathcal{F}_{t-1}\right) \\
&\leq \liminf_{N \to \infty} \mathbb{E}\left(M_t^{(N)} \mid \mathcal{F}_{t-1}\right) \\
&\leq \liminf_{N \to \infty} M_{t-1}^{(N)} \\
&= M_{t-1},
\end{aligned}
$$

which proves the first part of the claim. The second part of the claim follows from applying Fact 4 to the defined nonnegative supermartingale and rearranging. See Appendix B for details. ∎

The following corollary specializes Theorem 1 (and thus Corollary 3.5 of Abbasi-Yadkori [1]) to the case where $H$ is a RKHS and $f_t = k(\cdot, X_t)$, for all $t \geq 1$. In this special case, we can reframe the above theorem in terms familiar Gram matrix $K_t$, assuming the quantity is invertible. While we prefer the simplicity and elegance of working directly in the RKHS $H$ in the sequel, the follow corollary allows us to present Theorem 1 in a way that is computationally tractable.

**Corollary 1.** *Let us assume the same setup as Theorem 1, and additionally assume that (a)* $(H, \langle \cdot, \cdot \rangle_H)$ *is a RKHS associated with some kernel* $k$, *and (b) there is some* $\mathcal{X}$-*valued* $(\mathcal{F}_t)_{t \geq 0}$-*predictable process* $(X_t)_{t \geq 1}$ *such that* $(f_t)_{t \geq 1} = (k(\cdot, X_t))_{t \geq 1}$. *Then, for any* $\rho > 0$ *and* $\delta \in (0, 1)$, *we have that, with probability at least* $1 - \delta$, *simultaneously for all* $t \geq 0$,

$$\left\| (V_t + \rho \, \mathrm{id}_H)^{-1/2} S_t \right\|_H \leq \sigma \sqrt{2 \log \left( \sqrt{\frac{1}{\delta} \det(I_t + \rho^{-1} K_t)} \right)}.$$

*If, in addition, the Gram matrix* $K_t = (k(X_i, X_j))_{i,j \in [t]}$ *is invertible, we have the equality*

$$\|(I_t + \rho K_t^{-1})^{-1/2} \epsilon_{1:t}\|_2 = \|(\rho \, \mathrm{id}_H + V_t)^{-1/2} S_t\|_H.$$

We prove Corollary 1 in Appendix B. With this reframing of Theorem 1, we compare the concentration results of Theorem 1 (and thus Abbasi-Yadkori [1]) to the following, commonly leveraged result from Chowdhury and Gopalan [5].

**Fact 6** (**Theorem 1 from Chowdhury and Gopalan [5]**). *Assume the same setup as Fact 5. Let $\eta > 0$ be arbitrary, and let $K_t := (k(X_i, X_j))_{i,j \in [t]}$ be the Gram matrix corresponding to observations made by time $t \geq 1$. Then, with probability at least $1 - \delta$, simultaneously for all $t \geq 1$, we have*

$$\left\| \left( (K_t + \eta I_t)^{-1} + I_t \right)^{-1/2} \epsilon_{1:t} \right\|_2 \leq \sigma \sqrt{2 \log \left( \frac{1}{\delta} \sqrt{\det \left( (1 + \eta) I_t + K_t \right)} \right)}.$$

To make comparison with this bound clear, we parameterize the bounds in the above fact in terms of $\eta > 0$ instead of $\rho > 0$ to emphasize the following difference: both sides of the bound presented in Theorem 1 shrink as $\rho$ is increased, whereas both sides of the bound in Fact 6 increase as $\eta$ grows. Thus, increasing $\rho$ in Theorem 1 should be seen as decreasing $\eta$ in the bound of Chowdhury and Gopalan [5]. The bounds in Corollary 1 and Fact 6 coincide when $\rho = 1$ and $\eta \downarrow 0$ (per Lemma 1 in Chowdhury and Gopalan [5]), but are otherwise not equivalent for other choices of $\rho$ and $\eta$.

We believe Theorem 1 and Corollary 3.5 of Abbasi-Yadkori [1] to be signficantly more usable than the result of Chowdhury and Gopalan [5] for several reasons. First, the aforementioned bounds *directly* extend the method of mixtures (in particular, Fact 5) to potentially infinite-dimensional Hilbert spaces. This similarity in form allows us to leverage existing analysis of Abbasi-Yadkori et al. [2] to prove our regret bounds, with only slight modifications. This is in contrast to the more cumbersome regret analysis that leverages Fact 6, which is not only more difficult to follow, but also obtains inferior, sometimes super-linear regret guarantees.

Second, we note that Theorem 1 provides a bound that has a simple dependence on $\rho > 0$. In more detail, directly as a byproduct of the simplified bounds, Theorem 2 offers a regret bound that can readily be tuned in terms of $\rho$. Due to their use of a "double mixture" technique in proving Fact 6, Chowdhury and Gopalan [5] essentially wind up with a nested, doubly-regularized matrix $((K_t + \eta I_t)^{-1} + I_t)^{-1/2}$ with which they normalize the residuals $\epsilon_{1:t}$. In particular, this more complicated normalization make it difficult to understand how varying $\eta$ impacts regret guarantees, which we find to be essential for proving improved regret guarantees.

We note that the central bound discussed in this section *does not* provide an improvement in dependence on maximum information gain in the sense hypothesized by Vakili et al. [33]. In particular, the authors hypothesized the possibility of shaving a $\sqrt{\gamma_T}$ multiplicative factor off of self-normalized concentration inequalities in RKHS's. This was shown in a recent work (see Lattimore [18]) to be impossible in general. Instead, Theorem 1 and Corollary 3.5 of Abbasi-Yadkori [1] give one access to a family of bounds parameterized by the regularization parameter $\rho > 0$. As will be seen in the sequel, by optimizing over this parameter, one can obtain significant improvements in regret.

## 4 An Improved Regret Analysis of GP-UCB

In this section, we provide the second of our main contributions, which is an improved regret analysis for the GP-UCB algorithm. We provide a description of GP-UCB in Algorithm 1. While we state the algorithm directly in terms of quantities in the RKHS $H$, these quantities can be readily converted to those involving Gram matrices or Gaussian processes for those who prefer that perspective [5, 38].

As seen in Section 3, by carefully extending the "method of mixtures" technique (originally by Robbins) of Abbasi-Yadkori et al. [2], Abbasi-Yadkori [1] and de la Peña et al. [7, 8] to Hilbert spaces, we can construct self-normalized concentration inequalities that have simple dependence on the regularization parameter $\rho$. These simplified bounds, in conjunction with information about the eigendecay of the kernel $k$ [32], can be combined to carefully choose $\rho$ to obtain improved regret. We now present our main result.

**Theorem 2.** *Let $T > 0$ be a fixed time horizon, $\rho > 0$ a regularization parameter, and assume Assumptions 2 and 1 hold. Let $\delta \in (0, 1)$, and for $t \geq 1$ define*

$$U_t := \sigma \sqrt{2 \log \left( \frac{1}{\delta} \sqrt{\det(\mathrm{id}_H + \rho^{-1} V_t)} \right)} + \rho^{1/2} D.$$

---

**Algorithm 1** Gaussian Process Upper Confidence Bound (GP-UCB)

---

**Input:** Regularization parameter $\rho > 0$, norm bound $D$, confidence bounds $(U_t)_{t \geq 1}$, and time horizon $T$.

    Set $V_0 := 0$, $f_0 := 0$, $\mathcal{E}_0 := \{f \in H : \|f\|_H \leq D\}$

    **for** $t = 1, \ldots, T$ **do**

        Let $(X_t, \widetilde{f}_t) := \arg\max_{x \in \mathcal{X}, f \in \mathcal{E}_{t-1}} \langle f, k(\cdot, x) \rangle_H$

        Play action $X_t$ and observe reward $Y_t := f^*(X_t) + \epsilon_t$

        Set $V_t := V_{t-1} + k(\cdot, X_t)k(\cdot, X_t)^\top$ and $f_t := (V_t + \rho \mathrm{id}_H)^{-1} \Phi_t^\top Y_{1:t}$

        Set $\mathcal{E}_t := \left\{ f \in H : \left\| (V_t + \rho \mathrm{id}_H)^{1/2}(f_t - f) \right\|_H \leq U_t \right\}$

---

*Then, with probability at least $1 - \delta$, the regret of Algorithm 1 run with parameters $\rho, (U_t)_{t \geq 1}, D$ satisfies*

$$R_T = O\left( \gamma_T(\rho)\sqrt{T} + \sqrt{\rho \gamma_T(\rho) T} \right),$$

*where in the big-Oh notation above we treat $\delta, D, \sigma, B,$ and $L$ as being held constant. If the kernel $k$ experiences $(C, \beta)$-polynomial eigendecay for some $C > 0$ and $\beta > 1$, taking $\rho = O(T^{\frac{1}{1+\beta}})$ yields $R_T = \widetilde{O}\left( T^{\frac{3+\beta}{2+2\beta}} \right)^2$, which is always sub-linear in $T$.*

While we present the above bound with a fixed time-horizon, it can be made anytime by carefully applying a standard doubling argument (see Lattimore and Szepesvári [19], for instance). We specialize the above theorem to the case of the Matérn kernel in the following corollary.

**Corollary 2.** *Definition 3 states that the Matérn kernel with smoothness $\nu > 1/2$ in dimension $d$ experiences $(C, \frac{2\nu + d}{d})$-eigendecay, for some constnat $C > 0$. Thus, GP-UCB obtains a regret rate of $R_T = \widetilde{O}\left( T^{\frac{\nu + 2d}{2\nu + 2d}} \right)$.*

We note that our regret analysis is the first to show that GP-UCB attains sublinear regret for general kernels experiencing polynomial eigendecay. Of particular import is that Corollary 2 of Theorem 2 yields the first analysis of GP-UCB that implies sublinear regret for the Matérn kernel under general settings of ambient dimension $d$ and smoothness $\nu$. A recent result by Janz [16], using a uniform lengthscale argument, demonstrates that GP-UCB obtains sublinear regret for the specific case of the Matérn family when the parameter $\nu$ and dimension $d$ satisfy a uniform boundedness condition independent of scale. Our results are (a) more general, holding for *any* kernel exhibiting polynomial eigendecay, (b) don't require checking uniform boundedness independent of scale condition, and (c) follow from a simple regularization based argument. In particular, the arguments of Janz [16] require advanced functional analytic and Fourier analytic machinery.

We note that our analysis does not obtain optimal regret, as the theoretically interesting but computationally cumbersome SupKernelUCB algorithm [25, 34] obtains a slightly improved regret bound of $\widetilde{O}\left( T^{\frac{\beta+1}{2\beta}} \right)$ for $(C, \beta)$-polynomial eigendecay and $\widetilde{O}\left( T^{\frac{\nu+d}{2\nu+d}} \right)$ for the Matérn kernel with smoothness $\nu$ in dimension $d$. Due to the aforementioned result of Lattimore [18], which shows that improved dependence on maximum information gain cannot be generally obtained in Hilbert space concentration, we believe further improvements on regret analysis for GP-UCB may not possible.

To wrap up this section, we provide a proof sketch for Theorem 2. The entire proof, along with full statements and proofs of the technical lemmas, can be found in Appendix C.

***Proof Sketch for Theorem 2.*** Letting, for any $t \in [T]$, the "instantaneous regret" be defined as $r_t := f^*(x^*) - f^*(X_t)$, a standard argument yields that, with probability at least $1 - \delta$, simultaneously for all $t \in [T]$,

$$r_t \leq 2U_{t-1} \left\| (\rho \mathrm{id}_H + V_{t-1})^{-1/2} k(\cdot, X_t) \right\|_H.$$

---

[2]The notation $\widetilde{O}$ suppresses multiplicative, poly-logarithmic factors in $T$

A further standard argument using Cauchy-Schwarz and an elliptical potential argument yields

$$R_T = \sum_{t=1}^{T} r_t \le U_T \sqrt{2T \log \det(\mathrm{id}_H + \rho^{-1} V_T)}$$

$$= \left( \sigma \sqrt{2 \log \left( \frac{1}{\delta} \sqrt{\det(\mathrm{id}_H + \rho^{-1} V_T)} \right)} + \rho^{1/2} D \right) \sqrt{2T \log \det(\mathrm{id}_H + \rho^{-1} V_T)}$$

$$\le \left( \sigma \sqrt{2 \log(1/\delta)} + \sigma \sqrt{2 \gamma_T(\rho)} + \rho^{1/2} D \right) \sqrt{4T \gamma_T(\rho)} = O \left( \gamma_T(\rho) \sqrt{T} + \sqrt{\rho \gamma_T(\rho) T} \right),$$

which proves the first part of the claim. If, additionally, $k$ experiences $(C, \beta)$-polynomial eigendecay, we know that $\gamma_T(\rho) = \widetilde{O} \left( \left( \frac{T}{\rho} \right)^{1/\beta} \right)$ by Fact 2. Setting $\rho := O(T^{\frac{1}{1+\beta}})$ thus yields

$$R_T = O \left( \gamma_T(\rho) \sqrt{T} + \sqrt{\rho \gamma_T(\rho) T} \right) = \widetilde{O} \left( T^{\frac{3+\beta}{2+2\beta}} \right),$$

proving the second part of the claim.

∎

## 5 Conclusion

In this work, we present an improved analysis for the GP-UCB algorithm in the kernelized bandit problem. We provide the first analysis showing that GP-UCB obtains sublinear regret when the underlying kernel $k$ experiences polynomial eigendecay, which in particular implies sublinear regret rates for the practically relevant Matérn kernel. In particular, we show GP-UCB obtains regret $\widetilde{O} \left( T^{\frac{3+\beta}{2+2\beta}} \right)$ when $k$ experiences $(C, \beta)$-polynomial eigendecay, and regret $\widetilde{O} \left( T^{\frac{\nu+2d}{2\nu+2d}} \right)$ for the Matérn kernel with smoothness $\nu$ in dimension $d$.

Our contributions are twofold. First, we show the importance of finding the "right" concentration inequality for tackling problems in online learning — in this case the correct bound being a self-normalized inequality originally due to Abbasi-Yadkori [1]. We provide an independent proof of a result equivalent to Corollary 3.5 of Abbasi-Yadkori [1] in Theorem 1, and hope that our simplified, truncation-based analysis will make the result more accessible to researchers working on problems in kernelized learning. Second, we demonstrate the importance of regularization in the kernelized bandit problem. In particular, since the smoothness of the kernel $k$ governs the hardness of learning, by regularizing in proportion to the rate of eigendecay of $k$, one can obtain significantly improved regret bounds.

A shortcoming of our work is that, despite obtaining the first generally sublinear regret bounds for GP-UCB, our rates are not optimal. In particular, there are discretization-based algorithms, such as SupKernelUCB [34], which obtain slightly better regret bounds of $\widetilde{O} \left( T^{\frac{1+\beta}{2\beta}} \right)$ for $(C, \beta)$-polynomial eigendecay. We hypothesize that the vanilla GP-UCB algorithm, which involves constructing confidence ellipsoids directly in the RKHS $H$, cannot obtain this rate.

The common line of reasoning [33] is that because the Lin-UCB (the equivalent algorithm in $\mathbb{R}^d$) obtains the optimal regret rate of $\widetilde{O}(d\sqrt{T})$ in the linear bandit problem setting, then GP-UCB should attain optimal regret as well. In the linear bandit setting, there is no subtlety between estimating the optimal action and unknown slope vector, as these are one and the same. In the kernel bandit setting, estimating the function and optimal action are not equivalent tasks. In particular, the former serves in essence as a nuisance parameter in estimating the latter: tight estimation of unknown function under the Hilbert space norm implies tight estimation of the optimal action, but not the other way around. Existing optimal algorithms are successful because they discretize the input domain, which has finite metric dimension [26], and make no attempts to estimate the unknown function in RKHS norm. Since compact sets in RKHS's do not, in general, have finite metric dimension [35], this makes estimation of the unknown function a strictly more difficult task. In fact, recent work by Lattimore [18] demonstrate that self-normalized concentration in RKHS's, in general, cannot exhibit improved dependence on maximum information gain. This further supports our hypothesis on the further unimprovability of the regret analysis of GP-UCB past the improvements made in this paper.

# 6 Acknowledgements

AR acknowledges support from NSF DMS-2310718 and NSF IIS-2229881. ZSW and JW were supported in part by the NSF CNS2120667, a CyLab 2021 grant, a Google Faculty Research Award, and a Mozilla Research Grant. JW also acknowledges support from NSF GRFP grants DGE1745016 and DGE2140739.

We also would like to thank Xingyu Zhou and Johannes Kirschner for independently bringing to our attention the result from Abbasi-Yadkori [1] (Corollary 3.5) on self-normalized concentration in Hilbert spaces which is essentially equivalent to Theorem 1. We have rewritten the paper in a way that emphasizes the importance of this result and provides proper attribution to the original author.

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
