# OpenReview forum: "On the Sublinear Regret of GP-UCB"
_NeurIPS.cc/2023/Conference — NeurIPS 2023 poster_

### Official Review · Reviewer_uTye · 2023-06-21

**Soundness:** 3 good
**Presentation:** 3 good
**Contribution:** 2 fair
**Rating:** 4
**Confidence:** 4

**Summary:**

The paper proposes a new analysis for the popular GP-UCB algorithm. Under this new approach, it is shown that GP-UCB achieves a sub-linear regret (though not optimal), partially resolving the question of whether GP-UCB can achieve optimal regret.

**Strengths:**

The paper proposes a new self-normalized martingale type inequality for infinite dimensional Hilbert spaces, which might be of independent interest to the community. The sub-linear convergence of GP-UCB shown by the paper is indeed of interest to better understand the observed performance of popular algorithm.

**Weaknesses:**

I can see that there is clearly certain improvement over the existing result. However, my concern is whether the result warrants a full publication. From what I understand, the important trick is to analyse in the Hilbert space representation, which allows for a tighter analysis and a cleaner dependence on the regularization parameter.

I don't think that the approach taken in Chowdhury and Gopalan is incompatible with the philosophy presented in this paper, although I do agree that reaching this conclusion (specifically dependence on the regularization parameter) might not be as easy.

So overall, I feel there is certainly _some_ novel contribution here, especially with regards to understanding the behaviour of GP-UCB. However, I don't think there is sufficient contribution for a full paper at NeurIPS. I feel the result is more appropriate for a shorter paper or a letter to a journal. I would encourage the authors to submit to such a venue.

**Questions:**

No particular question.

**Limitations:**

Yes.

---

> ### Author Rebuttal · Authors · 2023-08-10
>
> [*1. I can see that there is clearly certain improvement over the existing result. However, my concern is whether the result warrants a full publication. From what I understand, the important trick is to analyse in the Hilbert space representation, which allows for a tighter analysis and a cleaner dependence on the regularization parameter.*]
>
> We believe that our result warrants publication due to obtaining improved regret bounds on a problem that has been open for many years. In particular, we obtain the first sublinear regret bounds in a very general setting that is of practical significance (i.e. the kernels for which our bounds apply are regularly used in ML/Data science in practice).
>
> [*2. I don't think that the approach taken in Chowdhury and Gopalan is incompatible with the philosophy presented in this paper, although I do agree that reaching this conclusion (specifically dependence on the regularization parameter) might not be as easy. *]
>
> To the best of our knowledge, the inequality constructed by Chowdhury and Gopalan cannot be used to achieve the same regret guarantee (at least non-trivially). Here is a heuristic justification. Increasing $\eta$ in the CG bound can be viewed as decreasing $\rho$ in our bound, with bounds coinciding when $\eta = 0$ and  $\rho = 1$, and being not equivalent for other values. We take $\rho = O(p(T))$, where $p(x) = x^\beta$ for some $\beta \in (0, 1/2)$. Since increasing $\rho$ in our bounds corresponds to decreasing $\eta$ in the CG bounds, and $p(T) >> 1$, we would need to select $\eta < 0$ to obtain a similar level of regularization, which is not permitted in the bound of CG, which stipulates $\eta > 0$. This is just a heuristic justification, of course, and perhaps with more effort the regret guarantees can be recovered from this bound, but in our opinion this is non-trivial and non-obvious. In our opinion, the presented concentration bound, which as some reviewers pointed out was obtained in Abbasi-Yadkori’s PhD thesis, is somehow the “right” concentration bound, while the Chowdhury-Gopalan bound is not, which then leads to a suboptimal regret bound.
>
>
>
> [*3. So overall, I feel there is certainly some novel contribution here, especially with regards to understanding the behaviour of GP-UCB. However, I don't think there is sufficient contribution for a full paper at NeurIPS. I feel the result is more appropriate for a shorter paper or a letter to a journal. I would encourage the authors to submit to such a venue.*]
>
> We believe the improved understanding of GP-UCB is (by itself) worthy as a contribution to the Neurips community, given that there were basic questions about its frequentist regret bound that were open but the original paper received a test-of-time award a few years ago. In particular, this algorithm has been investigated in many papers, but none has obtained sublinear regret for general kernels experiencing eigendecay. We find that the simplicity of our results is a feature of our results, as they are straightforward and easy for a reader with familiarity on the topic to parse. That is, a result can be important and useful without having a long proof or complicated analysis (and in fact, the simple proof can be more useful than a complex one because it can be adapted to other settings by followup work).

---

> > ### Comment · Reviewer_uTye · 2023-08-11
> > **Response to Rebuttal**
> >
> > Thank you for your response.
> >
> > As mentioned before, I agree that the contribution is novel and technique of trading off regularization with effective dimension is indeed neat. After reading the other reviewer comments, I feel that my estimate of the relevance to the NeurIPS community might have been misplaced. At the same time, based on the comments by Reviewer x2D4 (and certain references that I was unaware of) and the follow-up discussion, I feel that a major revision of the paper would better serve both the authors and the community.
> >
> > As a result, I would like to keep my score. I do encourage the authors to resubmit as this paper has interesting results and seems to be relevant to community (definitely more so than I initially estimated).

---

> > > ### Author Response · Authors · 2023-08-17
> > >
> > > *1. Thank you for your response. As mentioned before, I agree that the contribution is novel and technique of trading off regularization with effective dimension is indeed neat. After reading the other reviewer comments, I feel that my estimate of the relevance to the NeurIPS community might have been misplaced. At the same time, based on the comments by Reviewer x2D4 (and certain references that I was unaware of) and the follow-up discussion, I feel that a major revision of the paper would better serve both the authors and the community. As a result, I would like to keep my score. I do encourage the authors to resubmit as this paper has interesting results and seems to be relevant to community (definitely more so than I initially estimated).*
> > >
> > > We thank the reviewer for their follow up to our rebuttal, and agree that the result is relevant to the Neurips community. We have fully revised the manuscript to reflect the promised changes, but respect the reviewer’s choice to maintain their initial score.

---

### Official Review · Reviewer_x2D4 · 2023-06-30

**Soundness:** 4 excellent
**Presentation:** 1 poor
**Contribution:** 1 poor
**Rating:** 1
**Confidence:** 5

**Summary:**

This is a very well written paper with nice, clean results on self-normalised concentration for seperable RKHS. The paper claims two contributions:
1. A new self-normalised concentration result for seperable RKHSs.
2. Improving the regret of GP-UCB by tuning the regularisation parameter.



**Strengths:**

The write-up is of excellent quality. The bounds derived are very neat and strong. The trick used to get better regret is neat and easy to use.

**Weaknesses:**

The result claimed in point 1 is actually well known, and has been for over a decade. See Theorem 4.1 in [1]. Indeed, the authors of the submitted manuscript state: "To the best of our knowledge, Fact 6 is the only existing result on self-normalised, time-uniform concentration in RKHS's [sic]", where Fact 6 is from [2], and Fact 6 is implied by (and strictly weaker than) Theorem 4.1 of [1]. This is unfortunate, and completely not the authors fault ([2] should not have been accepted in the first place). However, this isn't an issue that can be resolved by adding a citation to [1] in a camera ready version: even the paper's title is, after all, "Improved Self-Normalized Concentration in Hilbert Spaces". For this reason, I recommend that the paper is rejected.

The result claimed in point 2 relies on trading off the effective dimension term versus the regularisation term. This is a neat idea and very much ought to be published, by itself, in a paper that doesn't claim 1 as a contribution. Point 2 is very similar to Theorem 31 and 32 of [3]. There, the effective dimension is likewise traded off, but with RKHS norm (which the regularisation multiplies). Theorem 31 uses this trick to achieve a weaker result (the resulting implied regret is not sublinear for all $\nu, d$ for the Mat\'ern kernel); under an additional assumption, which has only been established for a particular choice of $d, \nu$ (uniform boundedness of kernel eigenfunctions). Theorem 32 would give the same result for the Mat\'ern kernel as in the submitted manuscript. There, the trade-off is done by changing lengthscale---doing it via regularisation, as done in the submission, is a much better idea.

A rewrite of this paper that makes clear what is previous work and what is the authors contribution would warrant a strong accept.

[1] Abbasi-Yadkori, Yasin. Online learning for linearly parametrized control problems. Diss. 2013.

[2] Chowdhury, Sayak Ray, and Aditya Gopalan. "On kernelized multi-armed bandits." International Conference on Machine Learning. PMLR, 2017.

[3] Janz, David. Sequential decision making with feature-linear models. Diss. 2022.

**Questions:**

None

---

> ### Author Rebuttal · Authors · 2023-08-10
>
> [*1. The result claimed in point 1 is actually well known, and has been for over a decade. See Theorem 4.1 in [1]. Indeed, the authors of the submitted manuscript state: "To the best of our knowledge, Fact 6 is the only existing result on self-normalised, time-uniform concentration in RKHS's [sic]", where Fact 6 is from [2], and Fact 6 is implied by (and strictly weaker than) Theorem 4.1 of [1]. This is unfortunate, and completely not the authors fault ([2] should not have been accepted in the first place). However, this isn't an issue that can be resolved by adding a citation to [1] in a camera ready version: even the paper's title is, after all, "Improved Self-Normalized Concentration in Hilbert Spaces". For this reason, I recommend that the paper is rejected.*]
>
> We thank the reviewer for bringing attention to this result, as we were unaware of the thesis upon submitting. Moreover, we are surprised it has not been mentioned in the existing literature (e.g. the Chowdhury/Gopalan paper or Vakili open problem paper on RKHS concentration). Upon inspection, we agree that the mentioned result is essentially equivalent to what we have presented (it seems the result holds under a slightly more general setting in which the shift matrix is non-diagonal and just of trace class, but our argument can be readily extended to this setting as well). While we believe our proof is of independent interest due to its simplicity and lack of reliance on advanced functional analytic results/tools, we believe it is no longer fair to claim this bound as a primary contribution. We still believe our regret bounds are of significant interest due to their improvement over key results in the literature. We have now shifted the structure of our paper to present this as a primary result.  We believe that by downgrading the claimed novelty of the concentration result and focusing on the novel regret guarantees, the paper still deserves to be accepted under the new title of “On the Sublinear Regret of GP-UCB”, with the contribution of the existing bound properly acknowledged.
>
>
>
> [*2. The result claimed in point 2 relies on trading off the effective dimension term versus the regularisation term. This is a neat idea and very much ought to be published, by itself, in a paper that doesn't claim 1 as a contribution. Point 2 is very similar to Theorem 31 and 32 of [3]. There, the effective dimension is likewise traded off, but with RKHS norm (which the regularisation multiplies). Theorem 31 uses this trick to achieve a weaker result (the resulting implied regret is not sublinear for all for the Mat'ern kernel); under an additional assumption, which has only been established for a particular choice of (uniform boundedness of kernel eigenfunctions). Theorem 32 would give the same result for the Mat'ern kernel as in the submitted manuscript. There, the trade-off is done by changing lengthscale---doing it via regularisation, as done in the submission, is a much better idea.*]
>
> We thank the reviewer for this reference, as the approach used is very interesting. We agree our presented result is more general, more robust to choice of kernel, and the argument is more approachable. We have appropriately added a citation to this thesis and provided a comparison in the main body.

---

> > ### Comment · Reviewer_x2D4 · 2023-08-10
> >
> > Thank you for your reply. It appears that we agree on the substance of my review, but (unsurprisingly) not on the outcome it warrants.
> >
> > **I will not adjust my score,** in that I stand by my opinion that without the possibility of a second round of review, this paper should not be accepted.
> >
> > At the same time, if the AC were to be satisfied with accepting the authors promise to do justice to previous work and significantly rewrite the paper (and on the condition that the title may be edited, as to which I am unsure), **my hypothetical score for such a rewritten paper would likely be an 8.**

---

> > ### Comment · Reviewer_x2D4 · 2023-08-10
> >
> > Also, I have just seen an arxiv version of this work cited elsewhere. For the sake of respect for Yasin's contribution, I would urge the authors to make haste in correcting the record on this on arxiv.

---

> > > ### Author Response · Authors · 2023-08-17
> > >
> > > *1. Thank you for your reply. It appears that we agree on the substance of my review, but (unsurprisingly) not on the outcome it warrants. I will not adjust my score, in that I stand by my opinion that without the possibility of a second round of review, this paper should not be accepted. At the same time, if the AC were to be satisfied with accepting the authors promise to do justice to previous work and significantly rewrite the paper (and on the condition that the title may be edited, as to which I am unsure), my hypothetical score for such a rewritten paper would likely be an 8.*
> > >
> > > We thank the reviewer greatly for their hypothetical score. We have already fully edited the manuscript to reflect the promised changes, and thus disagree that another review cycle is necessary. We nonetheless respect the reviewer’s decision to maintain their original score.
> > >
> > > *2. Also, I have just seen an arxiv version of this work cited elsewhere. For the sake of respect for Yasin's contribution, I would urge the authors to make haste in correcting the record on this on arxiv.*
> > >
> > > Arxiv has been correspondingly updated to reflect the promised changes.

---

### Official Review · Reviewer_GVZJ · 2023-07-06

**Soundness:** 3 good
**Presentation:** 2 fair
**Contribution:** 3 good
**Rating:** 6
**Confidence:** 2

**Summary:**

This paper addresses the kernelized bandit problem and focuses on improving the regret bounds of the Gaussian Process Upper Confidence Bound (GP-UCB) algorithm. The authors introduce novel techniques to achieve nearly optimal regret rates for GP-UCB. These improvements surpass previous state-of-the-art analyses and partially resolve an open problem posed by Vakili et al.


**Strengths:**

The paper suggest a new analysis the leads to improved regret bounds for the GP-UCB algorithm.

The technique used in the paper is based on a novel concentration inequality (Theorem 1) that may be of independent interest.
In addition, they demonstrate that the consideration of regularization based on kernel smoothness contribute to the nearly optimal regret rates as well.


**Weaknesses:**

The main weakness is perhaps clarity of presentation that renders the paper less accessible to a non-expert in the field.

For example, the authors refer to the Matern kernel as a commonly used kernel, yet to someone not as familiar with the specific literature it is harder to interpret the result. Perhaps consider providing a bit more background on it. (e.g., it was not mentioned in [4])
Why is it commonly used and what are trade-offs with other kernels?

Also consider including a table comparing the bounds with previous analysis, similar to Table 1 in [27], that would be helpful as well.


**Questions:**

see above

---

> ### Author Rebuttal · Authors · 2023-08-10
>
> [*1. For example, the authors refer to the Matern kernel as a commonly used kernel, yet to someone not as familiar with the specific literature it is harder to interpret the result. Perhaps consider providing a bit more background on it. (e.g., it was not mentioned in [4]) Why is it commonly used and what are trade-offs with other kernels?]
>
> The Matern kernel is used in practice due to its ability to capture and approximate ``rougher’’ functions than the typically used squared exponential kernel. We have added some more intuition to describe this kernel and aid the reader.
>
> [*2. Also consider including a table comparing the bounds with previous analysis, similar to Table 1 in [27], that would be helpful as well.*]
>
> We did not add a table as there are only three points of comparison for our paper — the previous regret bound for GP-UCB under polynomial eigendecay, our result, and the optimal regret bound. Per this comment we have more explicitly presented the comparison between these results to aid the reader.

---

### Official Review · Reviewer_e62U · 2023-07-24

**Soundness:** 3 good
**Presentation:** 2 fair
**Contribution:** 4 excellent
**Rating:** 4
**Confidence:** 5

**Summary:**

This paper re-investigates the upper bound analysis of the popular GP-UCB algorithm (in its vanilla version), focusing on the cumulative regret for Matern kernel (and more generally, kernels with polynomial decay).  The study reveals that the previous non-sub-linear regret bounds can be surpassed.

**Strengths:**

From the overall achieved result of this paper, I would say the authors have made a great contribution to this community, addressing an existing major concern that GP-UCB on Matern kernel fails to achieve sub-linear cumulative regret.

This finding would have a significant impact on tons of exisintg upper bounds based on IGP-UCB (Chowdhury & Gopalan, 2017), and its certain variants. The behind insights may also inspire future improvements, to see if this bound can be finally optimal that counteracts the lower bound in (Scarlett et.al, 2017).

**Weaknesses:**

However, I have some reservations about their technical details, particularly regarding their derivation of Theorem 1. Here are several reasons:
* The technical contribution of this paper appears to be non-significant, as it essentially extends a well-known result from linear bandits (Abbasi-Yadkori et.at, 2011), by fusing several exsiting results/facts.
* The extension starts with assuming a finite-dimensional RKHS (with dimension N) and then somehow lets $N\to\infty$.
* Though the truncation technique is widely used in analyzing infinite-dimensional spaces, it usually involves rigorous analysis of the residual term, which is not seeing to be well-treated in this paper.
* In Appendix, line 461, the statement "there exists some..." seems to compress a lot of information.  While I understand the existence of such $N_t$ should be due to eigenvalue decays, such existence should be proven or more thoroughly discussed.
* It is confusing to write  $||A||_H \overrightarrow{dim(H)\to\infty} ||A||_H$, as the two RKHS spaces are essentially not the same (finite v.s. infinite), where they are both being denoted by the same $H$.
* Overall, I was expected to see a more formal proof by properly handling the residual error term, but what I see is more like "spamming" $N\to\infty$.

On the other hand, the presentation is not appealing due to:
* Section 1 lacks intuitions about the tools/ideas to be used, which is essential for a technical paper.
* Section 2 is too long, and could be significantly reduced either by moving some less-important content to Appendix, or just providing informal results.
* Some terminologies, like "double mixture," are not explained, leaving readers without clear understanding.

Miscellaneous minor issues:
* The templates seem to be from NeurIPS 2020, not 2023
* line 182: "We can then making..."
* Appendix line 471: "so so"
* What is $\rho \downarrow 0$?
* "Holder" should be Hölder
* No formal definition of RKHS norm?


**Questions:**

Since the self-normalized concentration can also be used to show bounds for GP-TS, as was done by (Chowdhury & Gopalan, 2017), does the improved Theorem 1 result in similar improved results for GP-TS?

---

> ### Author Rebuttal · Authors · 2023-08-10
>
> [*1. The technical contribution of this paper appears to be non-significant, as it essentially extends a well-known result from linear bandits (Abbasi-Yadkori et.at, 2011), by fusing several exsiting results/facts.*]
>
> We believe the simplicity of our arguments are what make the technical contributions significant. For instance, the fact that regularization can be used to greatly improve regret rates is intuitive and follows from stat/ML dogma, but has been largely ignored for this basic problem for over a decade. We show that applying a simple technique can actually make significant improvements on a long open ML problem.
>
> [*2. The extension starts with assuming a finite-dimensional RKHS (with dimension N) and then somehow lets /Though the truncation technique is widely used in analyzing infinite-dimensional spaces, it usually involves rigorous analysis of the residual term, which is not seeing to be well-treated in this paper. It is confusing to write, as the two RKHS spaces are essentially not the same (finite v.s. infinite), where they are both being denoted by the same./Overall, I was expected to see a more formal proof by properly handling the residual error term, but what I see is more like "spamming"*]
>
> We formally and rigorously show that such a limiting argument holds in the appendix, including the justification of the existence of all limits. Moreover, such limiting notations are fully justified since they are just concerning the convergence of sequences of scalars (i.e. it is not a problem that the dimensions of the spaces are changing). Due to our use of an analytic limiting argument, there is no need to explicitly bound error terms. We are happy to further discuss any questions with the proofs if the reviewer is not fully convinced, especially if there is a particular step in the proof that the reviewer does not understand or agree with, but barring a concrete objection the reviewer may have, we believe it is unfair to call our proof “spamming”.
>
> [*3. In Appendix, line 461, the statement "there exists some..." seems to compress a lot of information. While I understand the existence of such  should be due to eigenvalue decays, such existence should be proven or more thoroughly discussed.*]
>
> The existence of such $N$ does not depend on eigendecay, but rather on the assumption that the elements being summed (the $f_t$) have finite norm almost surely. We have made this more explicitly clear in the proof/theorem statement.
>
> [*4. Miscellaneous minor issues:*]
>
> We thank the reviewer for pointing out the miscellaneous issues and have added corrections/clarifications. In particular, the notation $\rho \downarrow 0$ means “the limit as $\rho$ is taken to monotonically decrease towards zero from above”.
>
> [*5. Section 1 lacks intuitions about the tools/ideas to be used, which is essential for a technical paper. Section 2 is too long, and could be significantly reduced either by moving some less-important content to Appendix, or just providing informal results. Some terminologies, like "double mixture," are not explained, leaving readers without clear understanding.*]
>
> We provide a brief description of our arguments (improved concentration and using simple regularization) in the contribution section of the paper and provide a high-level, intuitive sketch of our techniques in each of the main result sections (sections 3 and 4). We have now added an additional description of our technique to Section 1 for clarity. We believe the material discussed in Section 2 is essential for understanding our technique, results, and notation used throughout the main body of the paper. We have added a more detailed description of the double mixture technique to aid the reader in their understanding.
>
> [*6. Since the self-normalized concentration can also be used to show bounds for GP-TS, as was done by (Chowdhury & Gopalan, 2017), does the improved Theorem 1 result in similar improved results for GP-TS?*]
>
> We believe that our results can equally be applied to GP-TS, and leave it as future work to investigate this result.

---

> > ### Comment · Reviewer_e62U · 2023-08-11
> > **Response to Author Feedback**
> >
> > Thanks for the feedback.  Please see my latest response.
> >
> > Let's leave behind the questions regarding the first point, as it is no longer considered a contribution of this paper.   Forgive me that I was also not aware of the existence of Theorem 1, so that the previous assessment, which was based on regarding point 1 as a major contribution (though I questioned some of the technical details), is now invalid.  This leads me to conclude that, in its current state, the paper **needs a major revision**.
> >
> > While the idea behind the second point is intriguing, regrettably, I would only be able to recommend the acceptance of this paper if:
> > -   A significant amount of rewriting of the full paper.
> >
> > -   A deeper discussion/exploration of varying $\rho$ with $T$.  See the details below:
> >     -  Generally speaking, the idea of treating the regularization parameter $\rho$ as a variable is not new in Bayesian learning, e.g., in Baeysian quadrature (BQ), the optimal $\rho$ is related to the fill distance $h_X$, and in case of Matern-$\nu$ kernel, $\rho = h_X^{\nu}=\Theta(T^{-\frac{\nu}{d}})$.  For a concrete example, see Wynne’s paper “Convergence Guarantees for Gaussian Process Means With Misspecified Likelihoods and Smoothness”.
> >
> >     -  Given this context, I have actually contemplated a variable $\rho$ in a BO setting.  However, one common challenge is the lack of well-defined constraints on the upper limit of $\rho$. This is where this paper provides a unique perspective, by deriving a two-term upper bound that relies on $\rho$ and then equating the two. Consequently, **I acknowledge the paper's contribution** in clarifying a possible optimal value of $\rho$.
> >
> >     -  However, I have the following question that hope the authors can answer:  Since the regret improvement relies heavily on treating $\rho$ as a variable, specifically $\rho = T^{\frac{d}{2\nu+2d}}$ for Matern kernel.  I'm actually surprised to see that no order notation is included here, since the resulting $\rho$ would be excessively large either for theory or practice (and can grow with $T$...).  For example, if we set $T=1000$, $\nu=1.5$, $d=3$, we will get $\rho=10$.  As far as my understanding goes, GP-based algorithms would stop working if we ever set the regularization parameter to beyond, say $0.1$ or so.  As a counterexample to the BQ case I gave earlier, letting $\rho = \Theta(1000^{-1.5/3})\approx c*0.03$ appears to be much more reasonable.
> >
> >     - So, if your theory indicates that a larger $\rho$ can lead to a better regret bound, it would be interesting to see this experimentally.  Or is this because there is a mistake of not using $\rho=\Theta(T^{\frac{1}{1+\beta}})$?  In that case, you should indicate that the implied constant has to be polynomially small for the algorithm to actually work.
> >
> >  -  Conducting comprehensive experiments to support the theoretical finding in Theorem 2, which suggests that a larger value of $\rho$ results in an improved convergence rate.
> >
> > , which I believe any of these three can’t be done properly without another reviewing process.

---

> > > ### Author Response · Authors · 2023-08-17
> > >
> > > We thank the reviewer for their positive and constructive feedback.
> > >
> > > *1. While the idea behind the second point is intriguing, regrettably, I would only be able to recommend the acceptance of this paper if: A significant amount of rewriting of the full paper.*
> > >
> > > The paper has been rewritten to properly attribute the existing result, focusing on presenting just the regret bound as a contribution. We believe that this contribution on its own is significant enough to warrant an acceptance due to solving a long-standing multi-armed bandit problem, which is of significant importance in the theoretical ML community.
> > >
> > > *2. A deeper discussion/exploration of varying $\rho$ with $T$. See the details below:*
> > >
> > > We emphasize that our paper provides regret guarantees, which are typically provided up to absolute constants (i.e. constants independent of the time horizon $T$). Thus, to demonstrate the regret bound, it suffices to provide just a single, valid regularization parameter. In fact, taking $\rho = \Theta(T^{d/(2d + 2\nu)})$ will provide exactly the same regret guarantee. We have updated our manuscript to reflect the fact that *any* choice of $\rho$ proportional to chosen regularization parameter will be valid.
> > >
> > > Moreover, in terms of the earlier-provided counterexample of Wynne, the regularization parameters $\lambda_T$ chosen in the kernel least-squares objective (which the reviewer mentions are of the form $\lambda_T = O(T^{-p})$ for some power $p$) seem to be chosen with respect the normalized objective function $\frac{1}{T}\sum_{i = 1}^T (g(x_i) - y_i)^2 + \lambda_T \|g\|^2$, whereas the least-squares objective function used for the construction of our kernel ridge estimators is of the form $\sum_{i = 1}^T (g(x_i) - y_i)^2 + \lambda_T \| g\|^2$. Regularization parameters in the first objective can be mapped to parameters in the second objective via multiplying by $T$. Correspondingly, dividing our regularization parameter by the time-horizon $T$ would result in a parameter for the normalized objective function. Thus, due to the difference in the normalization of the objective function, we don’t see it as an issue that $\rho$ is chosen in direct proportion to (as opposed to inversely proportional to) the time horizon $T$.
> > >
> > >
> > > *3. Conducting comprehensive experiments to support the theoretical finding in Theorem, which suggests that a larger value of rho results in an improved convergence rate.*
> > >
> > > We believe that the results presented in this paper are of significant *theoretical* value, and thus we do not intend to include significant experimentation. The validity of the regret bounds follows directly from the validity of the arguments made in the proofs. The reviewer’s concerns about the magnitude of the regularization parameter are addressed in the preceding paragraph.

---

### Official Review · Reviewer_CiaU · 2023-07-27

**Soundness:** 4 excellent
**Presentation:** 4 excellent
**Contribution:** 2 fair
**Rating:** 6
**Confidence:** 4

**Summary:**

This paper addresses the open question of Vakili et al, on the possibility of obtaining rate-optimal regret bounds for kernelized bandits under the RKHS assumption. Authors present a slightly different RKHS martingale bound, which allows them to obtain a regret of the form $\rho\sqrt{\gamma_TT}+ \gamma_T\sqrt{T}$.  By carefully choosing the regularization parameter $\rho$ corresponding to the smoothness of the kernel, they are able to improve the rate of regret of the well known GP-UCB. This is particularly important in the case of rough Matern kernels, where the classic bounds become vacuous, while this paper still shows sublinearity.

**Strengths:**

1. The paper gives non-vacuous regret bounds for GP-UCB under RKHS assumption, despite many of the classical prior works (e.g. Srinivas  et al or Chowdhury & Gopalan).

2. I think the key strength is the nice and simple idea of tuning the regularization parameter according to the smoothness of the kernel. This choice makes a lot of sense and I am surprised that we have not been doing so all along.

3. The proof technique (truncation trick) for obtaining the $\mathcal H$-valued bound is also quite intuitive and interesting. I think the approach say in Lemma 2 may be used in other applications, to easily go from finite-dimensional analysis to kernels.

4. The paper is very well written, its clear and has a great flow. It gives good education!


**Weaknesses:**

1.This paper does not resolve the open question of Vakili et al. As far as I know, the key question within the field is to whether we can get a $\sqrt{T\gamma_T}$ bound on the regret in the RKHS setting, as we do in the GP setting.
Effectively, this paper improves the dependency of $\gamma_T$ on $T$, so that while we still have a $\gamma_T\sqrt{T}$ bound, the gap with the lower bound [Scarlet et al 2017] and the GP upper bound is reduced.

The authors conjecture in the conclusion that fully closing this gap may not be possible in the Conclusion section. I believe that is correct. In fact, the paper "A Lower Bound for Linear and Kernel Regression with Adaptive Covariates" [Lattimore, COLT 2023] addresses the same open problem, and shows that such confidence sets cannot be improved by any estimator.

2.I should also point out Chowdhurry & Gopalan is not the only result for bounding $\mathcal H$-valued martingales. As far as I know, this was first introduced in Theorem 3.11 in the PhD Thesis of Yasin Abbasi Yadkori in 2013, titled "Online Learning for Linearly Parametrized Control Problems" published by University of Alberta. Could you please compare the Theorem 1 in the paper with Theorem 3.11 of Abbasi-Yadkori? Due to difference in notation I can rigorously compare the bounds. How do the proof techniques differ?




**Questions:**

In Chowdhurry and Gopalan's paper, what would happen if we set $\eta$ also as a function of $\eta$?
Could we in the end obtain the same rate for regret? The bounding process (rhs of the inequality) in Theorem 1, is similar to that of Lemma 1 in Chowdhurry & Gopalan, while I can see that the martingale is slightly different.
Is this slight difference actually stopping them from getting the same rates? Or could it be that by tuning $\eta$ different than them (the simply set it to $1/t$) a similar rate as Cor 2 may be obtained?

**Limitations:**

Limitations are adequately addressed. The paper "A Lower Bound for Linear and Kernel Regression with Adaptive
Covariates" Lattimore COLT 2023 should be mentioned.

---

> ### Author Rebuttal · Authors · 2023-08-10
>
> We would like to thank the reviewer for the overall positive reviews and constructive feedback.
>
> [*1. This paper does not resolve the open question of Vakili et al. As far as I know, the key question within the field is to whether we can get a  bound on the regret in the RKHS setting, as we do in the GP setting.*]
>
> We agree that we do not derive confidence bounds that shave a $\sqrt{\gamma_T}$ factor off the existing bounds, but our bounds allow a user to optimize tightness (in terms of a regularization parameter) for a target time horizon. Importantly, we do address the question raised in the abstract of Vakili’s open problem paper of the fact that GP-UCB fails to obtain sublinear regret. We have now clarified our contributions towards resolving the open problem (that of showing improved regret bounds for GP-UCB, but not improving dependence on $sqrt{\gamma_T}$).
>
> [*2. The authors conjecture in the conclusion that fully closing this gap may not be possible in the Conclusion section. I believe that is correct. In fact, the paper "A Lower Bound for Linear and Kernel Regression with Adaptive Covariates" [Lattimore, COLT 2023] addresses the same open problem, and shows that such confidence sets cannot be improved by any estimator.*]
>
> We thank the reviewer for this reference. We have added a proper citation and discussion to the COLT’23 paper to address how this resolves the conjecture we posed.
>
> [*3. I should also point out Chowdhurry & Gopalan is not the only result for bounding H-valued martingales. As far as I know, this was first introduced in Theorem 3.11 in the PhD Thesis of Yasin Abbasi Yadkori in 2013, titled "Online Learning for Linearly Parametrized Control Problems" published by University of Alberta. Could you please compare the Theorem 1 in the paper with Theorem 3.11 of Abbasi-Yadkori? Due to difference in notation I can rigorously compare the bounds. How do the proof techniques differ?*]
>
> We thank the reviewer for this reference, as we were unaware of this reference upon submitting. Upon inspection, the bound presented in Abbasi-Yadkori’s thesis is equivalent to our bound (it is slightly more general in that it allows the shift matrix to not just be $\rho id_H$, but our result can be readily extended to this setting). We are surprised that this result has not been referenced by existing works in the literature, as it is neither mentioned in the Chowdhury Gopalan paper nor the Vakili paper. In line with reviewer x2D4, we will no longer claim the concentration result as a major contribution (though we will leave it in the paper’s appendix as a self-contained proof in our notation), but our regret analysis of GP-UCB regret bounds remains a novel and significant contribution. We have appropriately revised our work to reference this citation.
>
> In terms of the difference in proof techniques – our proof is a ground-up proof in the sense that it does not depend on the existence of certain classes of Gaussian measures (Lemmas 3.1 and 3.2) nor on advanced topological notions of convergence (e.g. weak operator topology convergence, Corollary 3.5) – only on elementary functional analytic techniques, like truncation-based arguments. We believe that our proof is more instructive, but we no longer believe it is a major contribution.
>
> [* 4. In Chowdhurry and Gopalan's paper, what would happen if we set also as a function of ? Could we in the end obtain the same rate for regret? The bounding process (rhs of the inequality) in Theorem 1, is similar to that of Lemma 1 in Chowdhurry & Gopalan, while I can see that the martingale is slightly different. Is this slight difference actually stopping them from getting the same rates? Or could it be that by tuning different than them (the simply set it to ) a similar rate as Cor 2 may be obtained?*]
>
> To the best of our knowledge, the inequality constructed by Chowdhury and Gopalan cannot be used to achieve the same regret guarantee (at least non-trivially). Here is a heuristic justification. Increasing $\eta$ in the CG bound can be viewed as decreasing $\rho$ in our bound, with bounds coinciding when $\eta = 0$ and  $\rho = 1$, and being not equivalent for other values. We take $\rho = O(p(T))$, where $p(x) = x^\beta$ for some $\beta \in (0, 1/2)$. Since increasing $\rho$ in our bounds corresponds to decreasing $\eta$ in the CG bounds, and $p(T) >> 1$, we would need to select $\eta < 0$ to obtain a similar level of regularization, which is not permitted in the bound of CG, which stipulates $\eta > 0$. This is just a heuristic justification, of course, and perhaps with more effort the regret guarantees can be recovered from this bound, but we failed to do so, and in our opinion this is non-trivial and non-obvious.

---

> > ### Comment · Reviewer_CiaU · 2023-08-10
> > **Response to Authors Rebuttal**
> >
> > Thank you for your response. I understand that the PhD Thesis of Abbasi-Yadkori is not a common read, and did not have necessarily expected the authors to be aware of it.
> >
> > Overall, since the authors have added the missing important references (and hopefully a discussion on them) I think this paper will be a valuable reference on H-valued Martingale bounds, and the proof techniques presented here have the potential of being used in other applications.

---

> > > ### Author Response · Authors · 2023-08-17
> > >
> > > We thank the reviewer for their support of our work. We indeed hope our work will be a useful reference for concentration in Hilbert spaces, and hope the simple idea of regularizing in proportion to the eigendecay of the underlying kernel will be applied in other bandit problems as well.

---

### Official Review · Reviewer_J83V · 2023-08-01

**Soundness:** 3 good
**Presentation:** 3 good
**Contribution:** 3 good
**Rating:** 7
**Confidence:** 3

**Summary:**

This paper investigates the performance of the Gaussian Process Upper Confidence Bound (GP-UCB) algorithm in kernelized bandit problems, focusing on the minimization of regret. The authors present a novel, self-normalized concentration inequality and a technique to regularize in proportion to the kernel's smoothness, which significantly simplifies the analysis of GP-UCB. The study reveals that GP-UCB, when properly regularized, provides nearly optimal, sublinear regret for kernels experiencing polynomial eigendecay, including the Matern kernel. The results validate the empirical performance of GP-UCB and emphasize the importance of thoughtful regularization in online learning problems.

**Strengths:**

NA

**Weaknesses:**

NA

**Questions:**

NA

---

> ### Author Rebuttal · Authors · 2023-08-10
>
> We would like to thank the reviewer for their overall positive reviews.

---

### Author Rebuttal · Authors · 2023-08-10

We would like to thank all reviewers for their thoughtful comments and feedback on our paper. We have responded to each of your comments, and we hope to address any follow-up questions you may have in the discussion.

Several reviewers have pointed out the existence of a bound in the thesis of Yasin Abbasi Yadkori that is equivalent to Theorem 1 of our work (the result on RKHS concentration). We were not aware of this result until recently, and are very surprised that this result has not been properly attributed in many central works on the kernel bandit problem. To address this, we have shifted the focus of our work to just being an improved regret analysis of GP-UCB (appropriately, we have retitled the paper “On the Sublinear Regret of GP-UCB”), with proper attribution to the result in Yasin’s thesis. We still include our independent proof of the bound in the appendix as we believe it to be of independent interest due to its simplicity. Given the overall positive feedback regarding the regret bounds (and the fact that our analysis resolves a question on the sub-linear regret of GP-UCB), we believe that our paper still warrants an accept.

---

### Decision · Program_Chairs · 2023-09-21

**Decision:**

Accept (poster)

**Comment:**

There has been quite a bit of discussion on the missed citation and thus incorrectly claiming the novelty. The authors acknowledged it and provided sufficient evidence and commitment for the revision for the final version. Since the paper still makes a solid contribution in improving the regret bound, I recommend an accept decision.